# Strong Evidence of the Role of Donkeys in the Epidemiology of *Leptospira* spp. in Semiarid Conditions

**DOI:** 10.3390/microorganisms11071853

**Published:** 2023-07-22

**Authors:** Davidianne de Andrade Morais, Bruno Cesar Nunes, Rafael Rodrigues Soares, Murilo Duarte de Oliveira, Diego Figueiredo da Costa, Hosaneide Gomes de Araújo, João Pessoa Araújo Júnior, Camila Dantas Malossi, Maria Luana Cristiny Rodrigues Silva, Sérgio Santos de Azevedo, Clebert José Alves

**Affiliations:** 1Centro de Saúde Tecnologia Rural (CSTR), Universidade Federal de Campina Grande (UFCG), Av. Universitária, s/n, Santa Cecília, Patos 58708-110, PB, Brazil; davidianneandrademorais@gmail.com (D.d.A.M.); nunesbcpernambuco@yahoo.com.br (B.C.N.); rafael_nh3@hotmail.com (R.R.S.); muriloduartevet@gmail.com (M.D.d.O.); hosaneidexavier@hotmail.com (H.G.d.A.); luacristiny@yahoo.com.br (M.L.C.R.S.); sergio.santos@professor.ufcg.edu.br (S.S.d.A.); 2Centro de Ciências Agrárias (CCA), Universidade Federal da Paraíba (UFPB), Rodovia BR 079, Km 02, Areia 58397-000, PB, Brazil; diegoveter@hotmail.com; 3Campus de Botucatu, Universidade Estadual Paulista (UNESP), Av. Prof. Mário Rubens Guimarães Montenegro, s/n, Botucatu 18618-687, SP, Brazil; joao.pessoa@unesp.br (J.P.A.J.); camilamalossi@gmail.com (C.D.M.)

**Keywords:** *Equidae*, One Health, leptospirosis, molecular detection, epidemiology, zoonosis

## Abstract

Donkeys (*Equus asinus*) are historically known for their close relationship to humanity, which raises the need to study zoonotic diseases that affect them. In this perspective, leptospirosis stands out as a disease with an economic and public health impact, and its occurrence is facilitated in times of higher rainfall indexes, especially in large urban centers. In view of the scarcity of information about leptospirosis in donkeys, the objective of this study was to detect the presence of *Leptospira* spp. and anti-leptospiral antibodies in donkeys rescued by a zoonosis center located in the Caatiga biome, Brazilian semiarid region. Overall, 30 donkeys of both sexes, aged between 4 months and 15 years, were used, from which 64 serum samples were collected and submitted to the microscopic agglutination test (MAT). In addition, 64 samples of urine, vaginal and preputial fluid, in duplicates, were subjected to the polymerase chain reaction (PCR) and microbiological. Sixteen (53.3%) animals tested positive in at least one diagnostic test, 12 (40%) of which were positive at MAT and seven (23.3%) in the molecular and bacteriological detection (urine, vaginal, and preputial fluid samples). This is the first report identifying donkeys infected with *Leptospira* spp. by molecular and bacteriological diagnosis in Brazil, and the first in the world to detect this agent in their genital fluids. The study also shows that donkeys are commonly exposed to leptospires in the Caatinga biome, and this constitutes a One Health-based concern, demonstrating the importance of broad studies where large numbers of humans and animals coexist when investigating zoonotic infections and when planning and implementing control measures for donkeys-associated leptospirosis.

## 1. Introduction

Leptospirosis is a cosmopolitan anthropozoonosis caused by bacteria of the genus *Leptospira* [1] which remains neglected and underreported even though it impacts the economy and public health. The agent is grouped into 30 serogroups and more than 300 serovars [2,3], each one with a preference for a particular host [4], and it is adapted to infect one or more animal species [5]. This sets up multiple transmission cycles, both in wild and anthropic environments [6]. Pathogenic *Leptospira* species have several hosts, and exposure ensues through direct contact with infected animals or indirectly via water and soil contaminated with urine. Transmission can also occur through contact with vaginal fluid and placental remains, copulation, and vertically. The losses in livestock production result from abortions, stillbirths, weak offspring, diminished growth rates, diminished milk production or agalactia, and death [1]. In addition, since the spectrum of hosts harboring and shedding leptospires from their renal tubules is broad and animals are often asymptomatic, the burden of leptospirosis has been underestimated and the ability to prevent infection and control animal diseases is lacking.

Donkeys (*Equus asinus*) are historically known for their close relationship with humans and can, therefore, act as important carriers of zoonotic agents. In addition, these animals are susceptible to infection by bacteria of the genus *Leptospira* and are naturally exposed to sources of infection [7,8]. Because donkeys are highly rustic animals with low zootechnical value, there is little concern for their health [9] as they are often subjected to precarious breeding environments, intense workloads, and abandonment when they decrease in performance or become ill. This scenario is commonly observed in semiarid regions and is of great health concern, particularly in urban environments. 

It is noteworthy that donkeys are of notable importance in the production of the mule species (*Equus mulus*), resulting from the mating between donkeys (*Equus asinus*) and horses (*Equus caballus*). In addition to being hybrids, these animals are of great rural importance due to their resistance and docility and therefore must also be free of contagious diseases, which is another justification for taking care of the health of donkeys, given the vertical transmission of leptospires. 

There are few studies that address *Leptospira* spp. infection in donkeys [10], especially when compared to the number of existing studies on infection in horses, even considering the zootechnical importance and the damage that leptospirosis can cause to donkeys. For diagnostic elucidation, laboratory tests are necessary, which can be direct with the identification of the agent, or indirect based on the detection of specific antibodies [11]. There is a scarcity of studies on the diagnosis of leptospirosis in donkeys, especially involving the direct detection of the agent in the species, with only two studies in this sense in the world literature [12,13]. This is of special importance in semiarid regions, such as the Caatinga biome in Brazil, which is a biome exclusive to Northeastern Brazil with abundant wild fauna—such as tamanduá-mirim (*Tamandua tetradactyla*), preá (*Cavia aperea*), mocó (*Kerodon rupestris*), and cachorro-do-mato (*Cerdocyon thous*), and offers unique epidemiological conditions that may influence the occurrence of infectious diseases such as leptospirosis. 

Therefore, the objective of this study was to detect the presence of *Leptospira* spp. and anti-leptospiral antibodies in donkeys rescued by a zoonosis center located in a large urban center in the Brazilian semiarid region through three biological material collections with an interval of one month between them. 

## 2. Material and Methods

### 2.1. Ethical Procedures

This project was approved by the Committee on Ethics in the Use of Animals of the Center for Health and Rural Technology at the Federal University of Campina Grande (CEUA/CSTR/UFCG), protocol number 50/2021.

### 2.2. Study Area and Animals

The study was conducted in a zoonoses center responsible for rescuing donkeys involved mainly in traction activities that have been abandonment in the urban areas of Campina Grande, the municipality with the second-largest population (413,830 inhabitants) in the state of Paraíba, Northeastern Brazil, located in the Caatinga biome, Brazilian semiarid. The region presents a tropical rainy climate (BSh) [14], with an altitude of 547.6 m above sea level, a maximum temperature of 32 °C and minimum of 19 °C, and average annual rainfall of 700 mm [15].

In April 2021, 30 animals (18 females and 12 males) were used, corresponding to the total number of donkeys present at the center, aged between 4 months and 15 years, fed with hay and with access to the public water supply. These animals were identified from numbers 1 to 30 through anklets placed on their hind limbs. Subsequently, samples were collected from these animals on two more occasions (one month between collections), totaling 20 animals in May (12 females and 8 males) and 14 in June (8 females and 6 males). At the end of the three months of collection, 64 samples of serum, urine, and genital fluid (vaginal/preputial) were obtained. At the time of sampling, a brief clinical evaluation was conducted on each of the animals through the measurement of their heart and respiratory rates, temperature, coloration of mucous membranes, and verification of the existence of ocular lesions. 

### 2.3. Biological Sample Collection

The blood samples were collected from the jugular vein using 8 mL labeled sterile tubes containing a coagulation activator after previous antisepsis of the puncture site with 2% iodine alcohol. After collection, the tubes were sent to the laboratory, where they were centrifuged at 1512× *g* for 10 min, and the serum samples were stored in microtubes at −20 °C.

Genital fluid samples directly from the cervicovaginal region and preputial ostium were collected with sterile swabs, after previous antisepsis of the external genital mucosa with 2% chlorhexidine digluconate (degerming solution). Then, after repeating the antisepsis of the external genital mucosa, in order to reduce the probability of contamination between the sites, urine samples were collected by spontaneous excretion, directly into sterile 15 mL Falcon tubes (Global Trade Technology, Jaboticabal, SP, Brazil).

### 2.4. Leptospira spp. Microbiological Isolation

For microbiological isolation, immediately after collection, 100 μL of genital fluid swab (vaginal/preputial) and urine were inoculated into separate tubes containing 5 mL of semisolid EMJH medium (Difco, BD Franklin Lakes, NJ, USA) enriched with amphotericin B (0.05 mg/mL), 5-fluorouracil (1 mg/mL), Fosfomycin (4 mg/mL), trimethoprim (0.2 mg/mL), and sulfamethoxazole (0.4 mg/mL) [16]. The tubes were stored in a biological oxygen demand incubator (BOD) at 28 °C for 24 h. After this period, they were subcultured in EMJH semisolid medium without antibiotics and stored in the BOD at 28 °C, being examined in a dark field microscope weekly for 12 weeks and replicated every 15 days to new EMJH semisolid medium without antibiotics.

### 2.5. Molecular Diagnosis

DNA extraction from urine, vaginal and preputial fluid samples, and the semi-solid EMJH medium culture was performed using the Dneasy Blood and Tissue Kit (Qiagen, Hilden, Germany) following the recommendations from the manufacturer. In the polymerase chain reaction (PCR), two primers [17] specific for pathogenic leptospires-*LipL*3245F (5′AAG CAT TAC CGC TTG TGG TG3′) and *LipL*32286R (5′GAA CTC CCA TTT CAG CGA TT3′) were used to amplify *LipL*32 gene. The methodology described by Hamond et al. [18] was followed, and primers were used in a concentration of 0.6 μM, 1.0 U Taq polymerase, 2.4 μM MgCl_2_, and 0.3 mM dNTP in a final volume of 25 μL. One cycle of initial denaturation at 94 °C for two minutes, followed by 35 cycles of denaturation at 94 °C for 30 s, annealing the primers to 53 °C for 30 s and a one-minute extension with 72 °C and final extension cycle at 72 °C for five minutes were used. PCR products were developed by 2% ultrapure agarose gel electrophoresis stained with Evans Blue (Thermo Fisher Scientific, Waltham, MA, USA) and 100 bp ladder, and DNA bands (≅260 bp) were visualized under ultraviolet light. Strain *Leptospira interrogans* serovar Copenhageni, Fiocruz L1-130 (ATCC BAA-1198) was used as a positive control, and ultrapure water was used as a negative control.

### 2.6. Serological Analysis

The presence of anti-*Leptospira* spp. antibodies was detected by the microscopic agglutination test (MAT) [19], using as antigens a living collection of 24 serovars from 18 serogroups, namely: *Leptospira biflexa*: Semaranga serogroup; *Leptospira interrogans*: Autumnalis, Australis, Bataviae, Canicola, Djasiman, Grippotyphosa, Sejroe, Pomona, Icterohaemorrhagiae, and Hebdomadis serogroups; *Leptospira borgpeterseni*: Ballum, Javanica, Mini, and Tarassovi serogroups; *Leptospira santarosai*: Shermani serogroup; *Leptospira noguchii*: Lousiana serogroup; and *Leptospira weilii*: Celledoni serogroup. The panel of antigens was provided by the Laboratory of Veterinary Bacteriology of the Fluminense Federal University, Niterói, Rio de Janeiro, Brazil, originating from the Pasteur Institute, France.

All samples with agglutinating activity at the 1:50 dilution (cut-off ≥ 50) were deemed positive [9], being serially titrated in a ratio of two. The antibody titer was the reciprocal of the highest dilution that showed 50% of agglutinations, and the highest titer achieved in each sample corresponded to the infecting serogroup.

## 3. Results

At clinical evaluation, all parameters were within normal limits, with no signs or symptoms that could be related to leptospirosis. Of the 30 animals analyzed, 16 (53.3%) were positive in at least one diagnostic method. Seven (23.3%) of these animals were positive in the molecular and bacteriological detection and 12 (40%) were positive in the MAT, with antibody titers ranging from 50 to 400. Three animals presented simultaneous positive reactions at MAT, PCR, and PCR of culture (Table 1). 

In the PCR, considering the 64 urine samples and the 64 samples of genital fluid collected throughout the trimester (April, May, and June 2021), leptospirotic DNA was verified in three (4.7%) urine samples (two males and one female) and six (9.4%) samples of genital fluid (four females and two males), with only one animal (animal 11) having a concomitant positive result in a urine sample and genital fluid, and only one animal (animal 1) having repeated positivity in two consecutive months (vaginal fluid in April and May). A total of 128 cultures were maintained in EMJH for 12 weeks, with leptospire growth in eight of these (three from urine, three from vaginal fluid, and two from preputial fluid), with DNA from *Leptospira* spp. detected in all of them, thus confirming the recovery of the agent in these cultures. 

Considering the 64 serological samples collected over the three months of research, overall seropositivity was observed in 19 (29.7%) of them, 11/30 (36.7%) in April, 6/20 (30%) in May, and 2/14 (14.3%) in June. The serogroups Icterohaemorrhagiae, Ballum, Canicola, Semaranga, and Grippotyphosa (Table 2) were identified. 

Persistent seropositivity was observed over the months for the same serogroup in five animals, three seropositive in April/May (animals 1, 14, and 22) and two in April/May/June (animals five and seven), all for the smallest titer, except animals seven and 14, which had a slight variation between 1:50 and 1:100. The highest titers observed (1:200 and 1:400) belonged to two reagent animals for the serogroup Icterohaemorrhagiae.

It was found that all three animals < 4 months of age (Table 1) were positive for at least one of the diagnostic tests, and two of them had positive results at PCR and PCR of culture of urine or preputial fluid. All seropositivities that were repeated over the months had relatively low titers (between 1:50 and 1:100) and were from animals ≥ 5 years of age.

## 4. Discussion

This is the first report of the identification of donkeys infected with *Leptospira* spp. by molecular and bacteriological diagnosis in Brazil and the first detection of the agent in genital fluids in this species in the world. In addition, most of the information available on leptospirosis in these animals comes from reports of seroprevalence and/or analogy to animals of the *Equidae* family. Because of this, crucial aspects of understanding the infection in donkeys are still not well described.

In molecular tests and bacteriological cultures, it was possible to detect, respectively, DNA and viable leptospires in samples of urine and genital fluid from males and females, indicating that venereal transmission can occur in both directions. In addition, the fact that most of these donkeys did not show concomitance results between the two sites surveyed may indicate a dissociation between them, highlighting the importance of the venereal route in the maintenance of leptospires and corroborating research findings in other species that genital leptospirosis is not a consequence of renal colonization, but which strains may preferentially colonize the genital tract [20,21,22,23,24].

Another finding that may strengthen the hypothesis of independence between urinary and genital sites in colonization by leptospires concerns positivity in PCR and cultures of urine and preputial fluid samples from immature animals (<4 months). Of these, the donkey that presented DNA in the urine did not present it in the preputial fluid, and the opposite happened with another animal. Therefore, it was evidenced that leptospires can be found in the male genital tracts of donkeys regardless of their sexual maturity.

Leptospires can be shed in semen and vaginal secretions, and transmitted by copulation or artificial insemination [25]. However, there was no evidence about this in donkeys until now. In addition, this route may contribute to the incidence of the infection by leptospires, since environmental factors do not influence it much [24,26]. In young animals, vertical transmission through milk or transplacental transmission can also be considered.

The positivity in the PCR and seronegativity in the MAT presented by animals 8, 11, 13, and 19 can be explained by the ability to detect bacterial DNA in the early stages of the infection before the development of a serological response to the infection [23,27,28], or in the chronic stage of the disease in domestic animals [3]. Furthermore, these animals could be seropositive to other serogroups, not included in the MAT panel of this study, or be attributed to a short period of seroconversion [16,29].

The opposite was observed in animals 4, 5, 7, 12, 14, 16, 18, 22, and 30, which were negative in the PCR and seropositive in the MAT. This finding may be related to concentrations of DNA from *Leptospira* spp. being below the detection threshold in the PCR, which is likely for chronic carriers [22,30]. Leptospire shedding in the urine is intermittent, therefore, even in cases of negative results in the PCR or culture, the possibility that the animal is a carrier cannot be excluded [30].

The frequency of seropositivity found in the MAT in this study is close to the results from other donkey populations in semiarid conditions, such as 28.6% in Brejo Paraibano, Brazil [5], 21.4% in the Sertão of Paraíba, Brazil [31], and 19.8% in the Sertão of Pernambuco, Brazil [9]. In the scientific literature, there are reports of seropositive donkeys for *Leptospira* spp. in Africa, America, Asia, and Europe, with frequencies varying from 7.7% [32] to 85% [7], and these variations may be associated with regional differences and risk factors. 

Most serological studies on leptospirosis use a cutoff point of 100. However, the adoption of the 1:50 cut−off point of the MAT allowed an increase in the detection of anti−leptospiral antibodies in the animals studied, including those that presented repeated seropositivity over the months, or that were positive in the molecular and bacteriological tests, revealing a possible standard cutoff point for this species. It is noteworthy that a seronegative animal is not always free of infection and the cut−off 50 has shown better performance in semi−arid regions.

All the repeated seropositivity over the months had relatively low titers (between 1:50 and 1:100) and were from older animals (≥5 years old), possibly due to being chronic carriers or having an infection associated with an immune system deficiency, which is commonly found in these animals [5]. Rocha et al. [33] suggested a possible development of immunity through early exposure and re−exposure to the agent that may, ultimately, result in some tolerance and residual titers in older horses.

The most frequent serogroups in this study, Icterohaemorrhagiae and Ballum, have synanthropic rodents as their main reservoirs [3,34,35] and are of relevant importance in human leptospirosis [36,37]. Other serogroups found were Grippotyphosa, which is related to rodents or other wild animals [5,8,38,39]; Canicol, which has dogs as reservoirs [10,35,40]; and Semaranga, a saprophytic strain associated with cross−reactions involving many infectious serovars, often absent from the panel of serogroups used in the MAT [41,42].

Thus, considering the serological findings, it is reasonable to assume that the exposure of donkeys to leptospires in this study may have occurred through contact with rodents, wild animals, and dogs, or contaminated water or food [6]. In addition, donkeys raised in urban areas are generally maintained in peripheral areas of the municipality, often urbanized due to a recent invasion of wild environments [43], bringing domestic animals and humans closer to reservoirs of *Leptospira* spp. belonging to the fauna.

It was observed that the newly rescued animals did not undergo a quarantine period before entering the zoonoses center. However, this did not prove to be a determining factor for the occurrence of infection, since 15 of the 16 animals that were positive in at least one of the tests showed anti−leptospiral antibodies or *Leptospira* spp. DNA in their first collection. In addition, the only animal that tested positive only after the second collection was seropositive for the Grippotyphosa serogroup, which was unprecedented concerning the others in this study. This may be a consequence of an infection acquired within the zoonosis center with a different serovar.

The results of this work indicate the elimination of leptospires in the urine and genital fluids of a significant number of donkeys tested, revealing a serious public health problem, in which these animals are sources of infection for the population in view of the circulation of donkeys in urban roads and their close relationship with humans. Therefore, these animals can act as important sentinels and contribute to local health authorities in mitigating risk factors shared between humans and animals [28,37].

It was not possible to maintain sampling uniformity until the end of the study because three animals died of undetermined causes shortly after the first collection and, in addition, 13 animals were adopted (seven in April and six in May). Another factor was the limited number of types of samples collected for detecting the agent, since live animals were used, raising the need to research *Leptospira* spp. in donkeys on a postmortem basis, to enable the analysis of a wider range of biological materials and better elucidation of the disease in the species.

With this study, it was possible to show the involvement of donkeys in the epidemiology of leptospirosis in semiarid conditions, in addition to unprecedented facts for the scientific community, which were the molecular and bacteriological detections of *Leptospira* spp. in genital fluids. These findings contribute to a better understanding of the role of this species as a carrier of leptospires, highlighting the concerns that these animals can cause in the One Health context.

## 5. Conclusions

In conclusion, it was possible to highlight the role of donkeys from the Caatinga biome, a semiarid region of Brazil, as potential carriers of *Leptospira* spp. through molecular and bacteriological identification of the agent. The study also shows that donkeys are commonly exposed to leptospires in the Caatinga biome, and this constitutes a One Health−based concern, demonstrating the importance of broad studies where large numbers of humans and animals coexist when investigating zoonotic infections and when planning and implementing control measures for donkeys−associated leptospirosis.

## Figures and Tables

**Table 1 microorganisms-11-01853-t001:** *Leptospira* spp. serological, molecular, and microbiological diagnostic results in donkeys from the Caatinga biome, Brazil, April to June 2021.

Animal Identification	Sex	Age	MAT				PCR		PCR of Culture	
			1:50	1:100	1:200	1:400	Urine	Fluid	Urine	Fluid
1 ^am^	F	8 years	+bal ^am^	−	−	−	−	+ ^am^	−	+ ^a^
2 ^amj^	F	1 year	−	−	−	−	−	−	−	−
3 ^amj^	M	3 years	−	−	−	−	−	−	−	−
4 ^a^	F	6 years	−	−	−	+ict ^a^	−	−	−	−
5 ^amj^	M	8 years	+bal ^am^	−	−	−	−	−	−	−
6 ^a^	M	9 years	+ict ^a^	−	−	−	+ ^a^	−	+ ^a^	−
7 ^amj^	M	8 years	+can ^a^	+can ^mj^	−	−	−	−	−	−
8 ^a^	M	<4 months	−	−	−	−	−	+ ^a^	−	+ ^a^
9 ^amj^	M	2 years	−	+gri ^m^	−	−	−	+ ^m^	−	+ ^m^
10 ^a^	M	9 years	−	−	−	−	−	−	−	−
11 ^a^	F	5 years	−	−	−	−	+ ^a^	+ ^a^	+ ^a^	+ ^a^
12 ^am^	F	7 years	−	−	+ict ^a^	−	−	−	−	−
13 ^amj^	M	<4 months	−	−	−	−	+ ^a^	−	+ ^a^	−
14 ^am^	F	10 years	+ict ^m^	+ict ^a^	−	−	−	−	−	−
15 ^a^	F	2 years	−	−	−	−	−	−	−	−
16 ^a^	F	8 years	+sem ^a^	−	−	−	−	−	−	−
17 ^amj^	F	3 years	−	−	−	−	−	−	−	−
18 ^a^	M	5 years	+sem ^a^	−	−	−	−	−	−	−
19 ^amj^	F	5 years	−	−	−	−	−	+ ^a^	−	+ ^a^
20 ^amj^	F	2 years	−	−	−	−	−	−	−	−
21 ^am^	F	2 years	−	−	−	−	−	−	−	−
22 ^am^	M	15 years	+ict ^am^	−	−	−	−	−	−	−
23 ^am^	M	2 years	−	−	−	−	−	−	−	−
24 ^amj^	F	1 year	−	−	−	−	−	−	−	−
25 ^amj^	F	4 years	−	−	−	−	−	−	−	−
26 ^amj^	M	9 years	−	−	−	−	−	−	−	−
27 ^amj^	F	5 years	−	−	−	−	−	−	−	−
28 ^amj^	F	2 years	−	−	−	−	−	−	−	−
29 ^a^	F	7 years	−	−	−	−	−	−	−	−
30 ^a^	F	<4 months	+sem ^a^	−	−	−	−	−	−	−

^a^ = sample collected in April; ^m^ = sample collected in May; ^am^ = sample collected on April and May; ^mj^ = sample collected in May and June; ^amj^ = sample collected in April, May and June; − = negative; + = positive; M = male; F = female; bal = Ballum; can = Canicola; ict = Icterohaemorrhagiae; gri = Grippotyphosa; sem = Semaranga.

**Table 2 microorganisms-11-01853-t002:** Most frequent serogroups of *Leptospira* spp. and respective antibody titers in donkeys from the Caatinga biome, Brazil, April to June 2021.

Month	Serogroup	Antibody Titers				
		1:50	1:100	1:200	1:400	Total (%)
April	Icterohaemorrhagiae	2	1	1	1	5 (45.4)
	Semaranga	3	0	0	0	3 (27.3)
	Ballum	2	0	0	0	2 (18.2)
	Canicola	1	0	0	0	1 (9.1)
	Total (%)	8 (72.7)	1 (9.1)	1 (9.1)	1 (9.1)	11 (100)
May	Ballum	2	0	0	0	2 (33.3)
	Canicola	0	1	0	0	1 (16.7)
	Icterohaemorrhagiae	2	0	0	0	2 (33.3)
	Grippotyphosa	0	1	0	0	1 (16.7)
	Total (%)	4 (66.7)	2 (33.3)	0	0	6 (100)
June	Ballum	1	0	0	0	1 (50)
	Canicola	0	1	0	0	1 (50)
	Total (%)	1 (50)	1 (50)	0	0	2 (100)
Trimester	Icterohaemorrhagiae	4	1	1	1	7 (36.8)
	Ballum	5	0	0	0	5 (26.3)
	Canicola	1	2	0	0	3 (15.8)
	Semaranga	3	0	0	0	3 (15.8)
	Grippotyphosa	0	1	0	0	1 (5.3)
	Total (%)	13 (68.4)	4 (21.1)	1 (5.3)	1 (5.3)	19 (100)

## Data Availability

Not applicable.

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
