# Peer review of "Strong Evidence of the Role of Donkeys in the Epidemiology of Leptospira spp. in Semiarid Conditions"

_microorganisms, 2023, doi:10.3390/microorganisms11071853_

Round 1

Reviewer 1 Report

In this study, the authors set out to determine the seropositivity and PCR positivity of urine and genital fluid in donkeys at a zoonosis center (is that really what it is--a zoonosis center--or is that a poor translation?). The number of donkeys evaluated was relatively small (30), but they had a surprisingly high rate of seropositivity and PCR positivity. Overall, the study is straightforward and accomplishes what it set out to do. This could have been improved with actual typing of the PCR result or PFGE of the cultures to identify the true infecting serovars in those donkeys.

Specific comments:

Page 2: “The male donkey breeds with a horse mare to produce mules and donkeys, …” Did the authors mean to write the sentence this way? I don’t know if the authors mean that donkey are used as work animals on farms because of their hardiness and stamina or whether they mean mules. This sentence also states “…and for this reason, these animals should be free from contagious disease.” Are the referring to the breeding or working? This sentence needs to be cleaned up.

Page 2: “Despite the significant number of donkeys…” This whole sentence is a bit run-on and seems to merge a variety of ideas into one poorly written sentence. This should be rewritten. Likewise, reference 10 is not a study on the “significant number of donkeys”, doesn’t demonstrate obvious damage to the donkeys (healthy donkeys had a higher seropositivity rate than ones with cutaneous sores or malnourished) and doesn’t really state that little is known (it does draw contrast to seropositive rates in donkeys from other studies).

Page 2: it’s fair to state that the wild fauna “may influence” but to state that it “can influence” is an overreach.

Page 6, last full paragraph: Although I generally agree with the authors, the lack of concurrent PCR positivity from both vaginal fluid and urine does not necessarily strengthen the hypothesis of independence. This is conjecture without any longitudinal study of the infection pattern of donkeys and no identification of the serovars identified on PCR. Had the one donkey with isolates from urine and genital fluid had 2 different serovars, that would have been fascinating. Or if all of the donkeys with urinary infection had a different isolate than genital, that would have also strengthened the hypothesis.

Page 7, 1st full paragraph: this is not surprising that apparently healthy animals would be seronegative and shedding leptospires. The same has been shown in dogs. Seropositivity is a poor predictor of shedding status in most animals.

Page 7, 2nd full paragraph, 4th sentence: “n” should be “in”.

Page 7, last full paragraph: Serology is a poor predictor of infecting serovar. Venereal transmission of leptospires would unlikely be associated with rodents. It would have been interesting to identify infecting serovars using MLST or other typing methods.

Page 7/8 paragraph: since serology is a poor predictor of infecting serogroup, the Grippotyphosa serovar in that donkey could be from infection at the zoonosis center with a different serovar.

Page 8, first paragraph, 4th sentence: I think the authors mean “rams” and not “names”.

Page 8, first paragraph, 7th sentence: it should read “fewer” not “less”.

Page 8, first paragraph, last sentence: the floating “e” should be “in”

The whole of the first paragraphs on Page 8 is poorly written. The first sentence does not make sense. This paragraph just does not seem like it was written by the same person who wrote the rest of the paper, or it wasn’t proofread very well after translation.

See above. Mostly the translation/English is fine, except for the few exceptions noted in the specific comments.

Author Response

Initially, I would like to thank the Reviewer for the valuable and productive comments, which were essential to improve the quality of the paper. All comments were addressed, and the changes are highlighted in yellow.

Prof. Dr. Sérgio Santos de Azevedo

COMMENTS AND RESPONSES

In this study, the authors set out to determine the seropositivity and PCR positivity of urine and genital fluid in donkeys at a zoonosis center (is that really what it is--a zoonosis center--or is that a poor translation?). The number of donkeys evaluated was relatively small (30), but they had a surprisingly high rate of seropositivity and PCR positivity. Overall, the study is straightforward and accomplishes what it set out to do. This could have been improved with actual typing of the PCR result or PFGE of the cultures to identify the true infecting serovars in those donkeys.

RESPONSE: Thanks for the valuable comments and recognition of the importance o four study. We strongly agree with the comment regarding the typing of the PCR result or PFGE of the cultures to identify the true infecting serovars, however, it was not possible to provide it at this time for financial reasons. It is noteworthy that we are planning a paper with molecular and serogrouping characterization of all isolates in this paper and other ones from other animals (cattle, sheep and goats). Thus, I kindly ask the Reviewer to reconsider our results as presented.

Specific comments:

Page 2: “The male donkey breeds with a horse mare to produce mules and donkeys, …” Did the authors mean to write the sentence this way? I don’t know if the authors mean that donkey are used as work animals on farms because of their hardiness and stamina or whether they mean mules. This sentence also states “…and for this reason, these animals should be free from contagious disease.” Are the referring to the breeding or working? This sentence needs to be cleaned up.

RESPONSE: Thanks for the comments. The text has been modified eccordingly: “It is noteworthy that donkeys are of notable importance in the production of the mule species (Equus mulus), resulting from the mating between donkeys (Equus asinus) and horses (Equus caballus). In addition to being hybrids, these animals are of great rural importance due to their resistance and docility and therefore must also be free of contagious diseases, which is another justification for taking care of the health of donkeys, given the vertical transmission of leptospires”.

Page 2: “Despite the significant number of donkeys…” This whole sentence is a bit run-on and seems to merge a variety of ideas into one poorly written sentence. This should be rewritten. Likewise, reference 10 is not a study on the “significant number of donkeys”, doesn’t demonstrate obvious damage to the donkeys (healthy donkeys had a higher seropositivity rate than ones with cutaneous sores or malnourished) and doesn’t really state that little is known (it does draw contrast to seropositive rates in donkeys from other studies).

RESPONSE: Thanks for the comments. The text has been modified eccordingly: “There are few studies that address Leptospira spp. infection in donkeys [10], especially when compared to the number of existing studies on infection in horses, even considering the zootechnical importance and the damage that leptospirosis can cause to donkeys”.

Page 2: it’s fair to state that the wild fauna “may influence” but to state that it “can influence” is an overreach.

RESPONSE: Thanks for the comments. The text has been modified eccordingly.

Page 6, last full paragraph: Although I generally agree with the authors, the lack of concurrent PCR positivity from both vaginal fluid and urine does not necessarily strengthen the hypothesis of independence. This is conjecture without any longitudinal study of the infection pattern of donkeys and no identification of the serovars identified on PCR. Had the one donkey with isolates from urine and genital fluid had 2 different serovars, that would have been fascinating. Or if all of the donkeys with urinary infection had a different isolate than genital, that would have also strengthened the hypothesis.

RESPONSE: Thanks for the valuable comments. We apologize for the wrong writing of the sentence. Our intention was to refer that the positivity in PCR and cultures of urine and preputial fluid samples from immature animals (<4 months) may strengthen the hypothesis of independence between urinary and genital sites in colonization by leptospires, as Bovine Genital Leptospirosis (BGL) has been recently proposed (https://doi.org/10.1016/j.theriogenology.2019.09.011) na there is a lacking of surveys on this subject in donkeys. The text has been modified accordingly: “Another finding that may strengthen the hypothesis of independence between urinary and genital sites in colonization by leptospires concerns positivity in PCR and cultures of urine and preputial fluid samples from immature animals (<4 months). Of these, the donkey that presented DNA in the urine did not present it in the preputial fluid, and the opposite happened with another animal. Therefore, it was evidenced that leptospires can be found in the male genital tract of donkeys regardless of their sexual maturity”.

Page 7, 1st full paragraph: this is not surprising that apparently healthy animals would be seronegative and shedding leptospires. The same has been shown in dogs. Seropositivity is a poor predictor of shedding status in most animals.

RESPONSE: Thanks for the valuable comments. We strongly agree with the comments. With the aforementioned paragraph we intend to emphasize that MAT is a poor predictor of carrier status in this species, especially in semiarid conditions.

Page 7, 2nd full paragraph, 4th sentence: “n” should be “in”.

RESPONSE: Thanks for the comments. The text has been modified eccordingly.

Page 7, last full paragraph: Serology is a poor predictor of infecting serovar. Venereal transmission of leptospires would unlikely be associated with rodents. It would have been interesting to identify infecting serovars using MLST or other typing methods.

RESPONSE: Thanks for the valuable comments. We strongly agree with the comments regarding the typing methods for identifying infecting serovars, such as MLST, however, it was not possible to provide it at this time for financial reasons. It is noteworthy that we are planning a paper with molecular and serogrouping characterization of all isolates in this paper and other ones from other animals (cattle, sheep and goats). Thus, I kindly ask the Reviewer to reconsider the text as presented.

Page 7/8 paragraph: since serology is a poor predictor of infecting serogroup, the Grippotyphosa serovar in that donkey could be from infection at the zoonosis center with a different serovar.

RESPONSE: Thanks for the comments. The information has been added: “In addition, the only animal that tested positive only after the second collection was seropositive for the Grippotyphosa serogroup, which was unprecedented concerning the others in this study. This may be a consequence of an infection acquired within the zoonosis center with a different serovar”.

Page 8, first paragraph, 4th sentence: I think the authors mean “rams” and not “names”.

RESPONSE: Thanks for the comments. The text has been modified eccordingly.

Page 8, first paragraph, 7th sentence: it should read “fewer” not “less”.

RESPONSE: Thanks for the comments. The text has been modified eccordingly.

Page 8, first paragraph, last sentence: the floating “e” should be “in”

RESPONSE: Thanks for the comments. The text has been modified eccordingly.

The whole of the first paragraphs on Page 8 is poorly written. The first sentence does not make sense. This paragraph just does not seem like it was written by the same person who wrote the rest of the paper, or it wasn’t proofread very well after translation.

RESPONSE: Thanks for the valuable comments. According to suggestion, the paragraphs have been modified, and some sentences have been deleted: “The results of this work indicate the elimination of leptospires in the urine and genital fluids of a significant number of donkeys tested, revealing a serious public health problem, in which these animals are sources of infection for the population in view of the circulation of donkeys in urban roads and their close relationship with humans. Therefore, these animals can act as important sentinels and contribute to local health authorities in mitigating risk factors shared between humans and animals [28,37].

               It was not possible to maintain sampling uniformity until the end of the study because three animals died of undetermined causes shortly after the first collection and, in addition, 13 animals were adopted (seven on April and six on May). Another factor was the limited number of types of samples collected for detecting the agent, since live animals were used, raising the need to research Leptospira spp. in donkeys on a post-mortem basis, to enable the analysis of a wider range of biological materials and better elucidation of the disease in the species.

               With this study, it was possible to show the involvement of donkeys in the epidemiology of leptospirosis in semiarid conditions, in addition to unprecedented facts for the scientific community, which were the molecular and bacteriological detections of Leptospira spp. in genital fluids. These findings contribute to a better understanding of the role of this species as a carrier of leptospires, highlighting the concerns that these animals can cause in the One Health context”.

Reviewer 2 Report

this study present an important contribution to the one health epidemiology studies on Leptospira in Brazil, including the coverage of a previously unscrutinised potential animal reservoir.

the study relies on its direct and simple methods that compliment each other and form a clear and wide-coverage response to the phenomenon, which becomes one of its main strengths.

I have, however, a few issues to raise to the authors, which I believe could be posteriorly reflected in the manuscript.

1. what is the relationship between the species/serovars, their method of detection, and their potential epitope (as in genital x kidney colonization). Are there relationships of the kind of "species x shows only in potential genital Leptospira colonization"? This could be interesting to highlight on the manuscript, as they could be pointing to specific ecological interactions displayed by the bacteria.

2. define "donkey". Are we talking about Equus asinus, all non-horse equids used for traction?

3. use the English vernacular names for the species that you cite during your introduction, mostly to facilitate identification for international readers. if you prefer, you can add the Brazilian Portuguese name in parenthesis together with the Latin binomial)

4. if you have the information, it would be interesting to plot the origin of the animals studied. their location of origin could be related to the species and serovar of Leptospira they carry.

5. what is the risk of urine contamination by genital Leptospira during sampling, given that you chose to collect it by regular micturition?

6. the second step of culture (the 12-week step on semisolid medium) was conducted under what conditions? please include in the methods

7. please, segregate your results by age as well, as it has been an aspect widely discussed in your manuscript

English is OK, there are a few typos and terms that are a bit awkward as part of the translation from a Romance language. a minor checkup by an independent third party would likely remediate the issue

Author Response

Initially, I would like to thank the Reviewer for the valuable and productive comments, which were essential to improve the quality of the paper. All comments were addressed, and the changes are highlighted in green.

Prof. Dr. Sérgio Santos de Azevedo

COMMENTS AND RESPONSES

This study present an important contribution to the one health epidemiology studies on Leptospira in Brazil, including the coverage of a previously unscrutinised potential animal reservoir.

The study relies on its direct and simple methods that compliment each other and form a clear and wide-coverage response to the phenomenon, which becomes one of its main strengths.

I have, however, a few issues to raise to the authors, which I believe could be posteriorly reflected in the manuscript.

RESPONSE: Thanks for the valuable comments and recognition of the importance o four study. All comments have been addressed accordingly.

  1. what is the relationship between the species/serovars, their method of detection, and their potential epitope (as in genital x kidney colonization). Are there relationships of the kind of "species x shows only in potential genital Leptospiracolonization"? This could be interesting to highlight on the manuscript, as they could be pointing to specific ecological interactions displayed by the bacteria.

RESPONSE: Thanks for the comments. Although we agree with the Reviewer that this type of analysis could be very important, in this case it would be impaired due to the number of animals used. Furthermore, considering the non-concomitance of serology results with molecular assays and the fact that serology provides information only on serogroups, where a single serogroup may include several species of leptospires, we consider that it would be uncertain to speak of predilection by tract based on serological results. Bovine Genital Leptospirosis (BGL) has been recently proposed (https://doi.org/10.1016/j.theriogenology.2019.09.011) na there is a lacking of surveys on this subject in donkeys, and our survey is an initial step to elucidate this possible syndrome in donkeys. Thus, I ask the Reviewer to consider our rebutal of not addressing conjectural issues. However, we are planning a paper with molecular and serogrouping characterization of all isolates in this paper and other ones from other animals (cattle, sheep and goats), and the suggestion by the Reviewer will be very valuable.

  1. define "donkey". Are we talking about Equus asinus, all non-horse equids used for traction?

RESPONSE: Thanks for the comments. We used Equus asinus, and this information has been added.

  1. use the English vernacular names for the species that you cite during your introduction, mostly to facilitate identification for international readers. if you prefer, you can add the Brazilian Portuguese name in parenthesis together with the Latin binomial)

RESPONSE: Thanks for the comments. The text has been modified eccordingly.

  1. if you have the information, it would be interesting to plot the origin of the animals studied. their location of origin could be related to the species and serovar of Leptospira they carry.

RESPONSE: Thanks for the comments. Unfortunately, this information was not possible to obtain because most animals were abandoned in the urban area, and it was not possible to identify their origin.

  1. what is the risk of urine contamination by genital Leptospira during sampling, given that you chose to collect it by regular micturition?

RESPONSE: Thanks for the comments. Information on previous antisepsis on the external genital mucosa was added. This procedure was adopted aiming to reduce the possible contamination between sites. The information has been added in the item 2.3: “Genital fluid samples directly from the cervicovaginal region and preputial ostium were collected with sterile swabs, after previous antisepsis of the external genital mucosa with 2% chlorhexidine digluconate (degerming solution). Then, after repeating the antisepsis of the external genital mucosa, in order to reduce the probability of contamination between the sites, urine samples were collected by spontaneous excretion, directly into sterile 15 mL Falcon tubes (Global Trade Technology, Jaboticabal, SP, Brazil)”.

  1. the second step of culture (the 12-week step on semisolid medium) was conducted under what conditions? please include in the methods

RESPONSE: Thanks for the comments. Information has been added to the text: “After this period, they were subcultured in EMJH semisolid medium without antibiotics and stored in the BOD at 28oC, being examined in a dark field microscope weekly for 12 weeks and replicated every 15 days to new EMJH semisolid medium without antibiotics”.

  1. please, segregate your results by age as well, as it has been an aspect widely discussed in your manuscript

RESPONSE: Thanks for the valuable comments. We have added the age information for each animal on Table 1, and a setence in the last paragraph of results: “It was found that all three animals < 4 months of age (Table 1) were positive for at least one of the diagnostic tests, and two of them had positive results at PCR and PCR of culture of urine or preputial fluid. All seropositivities that were repeated over the months had relatively low titers (between 1:50 and 1:100) and were from animals ≥ 5 years of age”.